# Rapid evaluation of bioactive Ti-based surfaces using an in vitro titration method

Weitian Zhao [1], David Michalik[2], Stephen Ferguson[3,4], Willy Hofstetter[5], Jacques Lemaître [1], Brigitte von Rechenberg[2,4] & Paul Bowen[1]

The prediction of implant behavior in vivo by the use of easy-to-perform in vitro methods is of great interest in biomaterials research. Simulated body fluids (SBFs) have been proposed and widely used to evaluate the bone-bonding ability of implant materials. In view of its limitations, we report here a rapid in vitro method based on calcium titration for the evaluation of in vivo bioactivity. Using four different titanium surfaces, this method identifies that alkaline treatment is the key process to confer bioactivity to titanium whereas no significant effect from heat treatment is observed. The presence of bioactive titanium surfaces in the solution during calcium titration induces an earlier nucleation of crystalline calcium phosphates and changes the crystallization pathway. The conclusions from this method are also supported by the standard SBF test (ISO 23317), in vitro cell culture tests using osteoblasts and in vivo animal experiments employing a pelvic sheep model.

[1] Institute of Materials, École Polytechnique Fédérale de Lausanne (EPFL), Lausanne, Switzerland. [2] Musculoskeletal Research Unit, University of Zürich, Zürich, Switzerland. [3] Institute for Biomechanics, ETH-Zürich, Zürich, Switzerland. [4] Competence Center for Applied Biotechnology and Molecular Medicine (CABMM), University of Zurich, Zurich, Switzerland. [5] Graduate School of Cellular and Biomedical Sciences, University of Bern, Bern, Switzerland. Correspondence and requests for materials should be addressed to W.Z. (email: weitian.zhao@epfl.ch)

Understanding biomineralization processes carries an important medical significance. As one of its applications, the in vivo deposited calcium phosphate layer on bone implants can help to diminish the typical adverse host response and facilitate the formation of a direct bonding with the bone[1–5]. Generally, the biocompatibility and/or the bioactivity study of medical implants is a critical component in biomaterials research and well-established in vitro methods are highly desirable compared with animal experiments for their economic and ethical advantages.

Previously, Kokubo proposed an in vitro method using simulated body fluids (SBFs) with ion concentration similar to biological fluids to test implant materials (ISO 23317)[6–8]. According to this method, materials that are able to induce hydroxyapatite formation on its surface after a 4-week immersion in the otherwise stable SBF are considered bioactive and would result in a better bonding with the bone after implantation[7]. This method is relatively simple to perform and is widely used in the research of bioglass and Ti-based materials[9–12].

However, despite its success and the following efforts to improve the current testing protocol[13], the method is fundamentally limited due to its inability to produce a quantitative measure for the bioactivity of a material. After a testing duration of 1–4 weeks, the bioactivity is simply categorized into "yes" (with apatite observed on material's surface) and "no" (no apatite observed). Comparison between different materials are difficult. Although the amount of precipitated hydroxyapatite can be quantified, it is often also a function of other factors, such as the available surface area. In fact, after the initial nucleation stage, the growth of hydroxyapatite crystals is dominated by secondary nucleation[14–16]. This contribution from secondary nucleation makes the isolation of the effect from surface chemistry a challenging task. In addition, although the method can easily distinguish inert materials (such as untreated titanium surface) and bioactive materials (such as bioglass), it struggles with materials having moderate activity. The use of a metastable solution for up to a month of testing duration as recommended in the ISO protocol amplifies the subtle changes during the test, sometimes producing inconsistent results[13,17,18]. Factors such as the vicinity to other surfaces in the static SBF test was also reported to influence the result[17,19], meaning that a material can be either inert or bioactive depending on slight differences in operational procedures.

In this paper, we try to address the above-mentioned issue and propose a different in vitro protocol based on a titration method. A calcium solution is continuously added to a phosphate solution containing the testing material until calcium phosphate nucleation takes place, as monitored by a calcium selective electrode and a pH electrode. The degree of supersaturation of the solution required for the nucleation event is used as an indication of the bioactivity of the material, i.e., the lower the supersaturation, the higher the bioactivity. Four titanium-based surfaces with different chemical treatments are tested. The solution behavior in the absence of a testing sample is first of all discussed and compared with the results from titrations in the presence of four surfaces. Using the same surfaces, this method is also directly compared with the standard static SBF test. To verify the proposed in vitro method, animal experiments using a sheep pelvic model are carried out. Experimental designs using a total of 96 implants in 8 sheep are performed followed by biomechanical tests and histological studies. Bone-implant-contact area as well as torque values after implantation are measured and used as an indication of osseointegration for the evaluation of the bioactivity of different implants. In addition, cell culture tests using osteoblasts are conducted. Both cell viability and biosynthetic activity as measured by alkaline phosphatase (ALP) activity are determined.

These studies aim to corroborate the proposed in vitro titration method for the evaluation of bioactive materials.

## Results

**Titration experiment without the presence of samples**. The product from the titration experiment was first determined in the absence of a testing material. Figure 1 shows the profile of free calcium ion ($Ca^{2+}$), added calcium, complexed and precipitated calcium concentration (the difference between added Ca and measured free $Ca^{2+}$) and the evolution of pH. The initial phosphate concentration of 4 mM leads to a partial complexation of $Ca^{2+}$ ions mostly in the form of $[CaHPO_4]^0$ according to existing thermodynamic models[20,21]. This can largely explain the gradual increase in the complexed and precipitated Ca before the onset point at ~180 min (reproducibly determined to be 185.6 ± 6.8 min from five independent experiments). From the curve, a clear two-step process can be observed, each corresponding to a sudden decrease of available free $Ca^{2+}$ and a distinct change of slope in the pH profile. The complexed and precipitated Ca continues to increase after these two events, indicating a continuous growth of the precipitated product. However, the consumption rate gradually slowed down probably due to a limited amount of phosphate counter ions. The evolution of the relative supersaturation during the titration experiment can be theoretically calculated (see below) and will be discussed in detail later.

The reaction products from different stages of the titration were sampled using a TEM grid and characterized using selected area electron diffraction (SAED). Figure 2 shows the morphology of the reaction product and the corresponding SAED patterns at two different time points: just before the second major event at near 250 min (Fig. 2c, d) and the final stage of steady increase in both free $Ca^{2+}$ and complexed and precipitated Ca at ~325 min (Fig. 2a, b). The normalized radial distribution density extracted from the SAED pattern is plotted in Fig. 2e. A commercial nanocrystalline hydroxyapatite was also analyzed in the same way and used as a reference pattern (morphology and SAED pattern given in Supplementary Fig. 1). The profile showing noncrystalline features is taken from the observed nanometer-sized species in the solution before the first major event at 180 min. Due to their small sizes (<5 nm) and the preparation procedures involved with TEM imaging, it is not certain that these species are actually present in the solution and their native morphologies

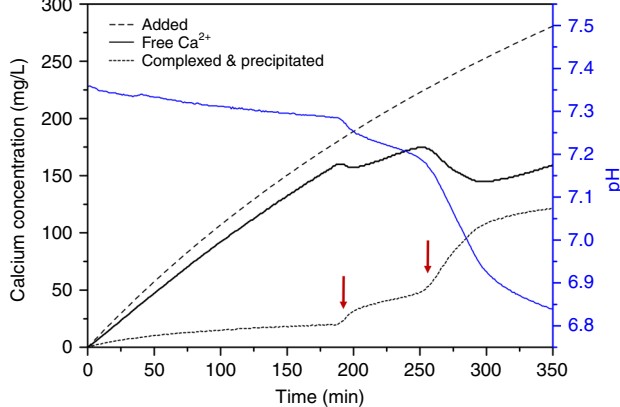

**Fig. 1** Free calcium profile of the titration system without external surfaces. The black line gives the free calcium concentration measured by a calcium ISE. The added amount of calcium ions is given by the dash line. The difference of the two indicates complexed and precipitated calcium, shown by the dotted line. Two red arrows indicate the onset point of two distinct events that resulted in the rapid consumption of free $Ca^{2+}$ ions. The pH is given by the blue line

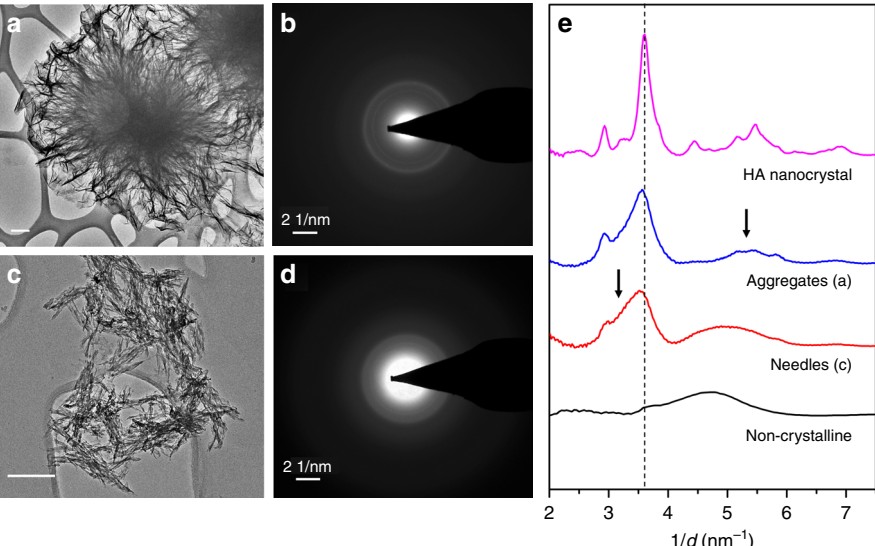

**Fig. 2** TEM images of products of the homogeneous nucleation from the titration experiment. **a** Final stage of titration near 325 min showing a morphology of aggregates of nano-sheets forming a globular shape, **b** is the corresponding SAED pattern. **c** Just before second major event near 250 min showing a needle-like crystal morphologies; **d** is the corresponding SAED pattern. Scale bars in the TEM images are 200 nm. **e** The normalized radial distribution density of the SAED patterns shown in **b** and **d** plus a commercial nanocrystalline HA (purple) and the nanometer-sized particles in the solution before 180 min (black) analyzed using the same method. The characteristic structural features in the SAED patterns are indicated with arrows

are not altered during conventional TEM sample preparation. The possibility of the existence of solution clusters is further discussed in the discussion section. However, from the analysis of the SAED pattern, it can be concluded that no crystalline species are present in the solution before the first event at ~180 min. The reaction product, the needle-shaped crystals, from the first event shows similar features as standard HA. However, the extra middle component between two peaks at ~3.0 and 3.6 nm$^{-1}$ as indicated by an arrow and the lower position of the main peak near 3.6 nm$^{-1}$ suggests a larger d-spacing that may better match octacalcium phosphate (OCP, $Ca_8(HPO_4)_2(PO_4)_4(H_2O)_5$, PDF #79-423) rather than HA ($Ca_5(PO_4)_3(OH)$, PDF #86-740), indicating a possible formation of OCP. However, the main feature in the SAED pattern that can distinguish OCP and HA occurs at very small angles that are masked by the beam stop. In addition, even if full spectra can be obtained, the small thickness of crystals might also result in the absence of this characteristic (100) peak of OCP which corresponds to planes with large d-spacing of 1.85 nm, as already reported in the literature[22]. After the second peak in the free calcium profile, the product exhibits a typical aggregated morphology adopted by HA (Fig. 2a). From the SAED pattern, it is also clear that the product has more crystalline features approaching that of commercial pure HA, including the emerging peaks at lower d-spacing (higher values in 1/d). A gradual structural transformation is expected after the second event. Considering HA is the thermodynamically most stable phase, this phase is most likely HA, and the presence of these crystals greatly promotes secondary nucleation and thus results in a drastic decrease in free calcium concentration.

**Titration experiment in the presence of samples**. Titanium powders with four different surface modifications were investigated in the titration experiment. Their behaviors after chemical treatments are similar to that reported for titanium discs[13]. Generally, heat treatment in air induced $TiO_2$ rutile formation on the surface, while NaOH treatment led to the formation of a porous surface layer rich in Ti, Na, and O, which was further transformed by heat treatment into $Na_2Ti_6O_{13}$, $TiO_2$ rutile and possibly a small amount of other sodium titanate phases

(Supplementary Fig. 2, 3). The porous amorphous layer after NaOH treatment is reported to be capable of ionic exchange with the solution medium. Therefore, separate experiments focusing on the decrease of solution free calcium from the four powders are carried out in different concentration ranges of calcium using similar solutions as in the titration but without the presence of phosphate to avoid CaP precipitation (details in Supplementary Discussion). The results show that 0.1 g NaOH-treated Ti (denoted hereafter as Ti NaOH) takes up (either on the surface or in the structure) 0.163 ± 0.004 mM free $Ca^{2+}$ ions when between 2 and 8 mL of calcium solution is added to a 50 mL buffered starting solution (total eight data points), corresponding to a concentration range of 0.77–2.76 mM Ca (26–134 min in the titration curve), indicating that the total uptake of calcium is rather constant after certain calcium concentration (Supplementary Table 1, Supplementary Fig. 4). On the contrary, for NaOH plus heat-treated Ti (denoted hereafter as Ti NaOH HT), a decrease of only 0.012 ± 0.002 mM of $Ca^{2+}$ (at 0.77 mM) was observed with two replicates, indicating that most exchangeable $Na^+$ ions are immobilized in the crystal lattice as various sodium titanate phases after heat treatment. Ti and Ti HT (heat-treated Ti) did not show detectable $Ca^{2+}$ adsorption with a detection limit of 0.1 mg/L. These results are at the same time supported by the features in the titration curves presented in Fig. 3 by focusing on the shape of the free calcium profile at early time points (0–20 min).

Figure 3 shows the titration curves for the four titanium surfaces: Ti, Ti HT, Ti NaOH, and Ti NaOH HT. It can be seen that the presence of Ti or Ti HT powder does not seem to have an effect on the CaP formation pathway in the solution as reflected in the shape of the curve. The two-step pathway is again followed with a distinct separation of the two events. Sampling the solution using a TEM grid at the second peak in the free calcium profile, crystals can be observed both on Ti powder surface as well as in the solution (Supplementary Fig. 5a). Similarly, at the end of the titration between 250 and 300 min, crystals with characteristic HA morphologies in the form of nano-sheets aggregated into globules can be seen in the solution (Supplementary Fig. 5b), an evidence of homogenous nucleation products from the solution. However, in the case of Ti NaOH and Ti NaOH HT powders,

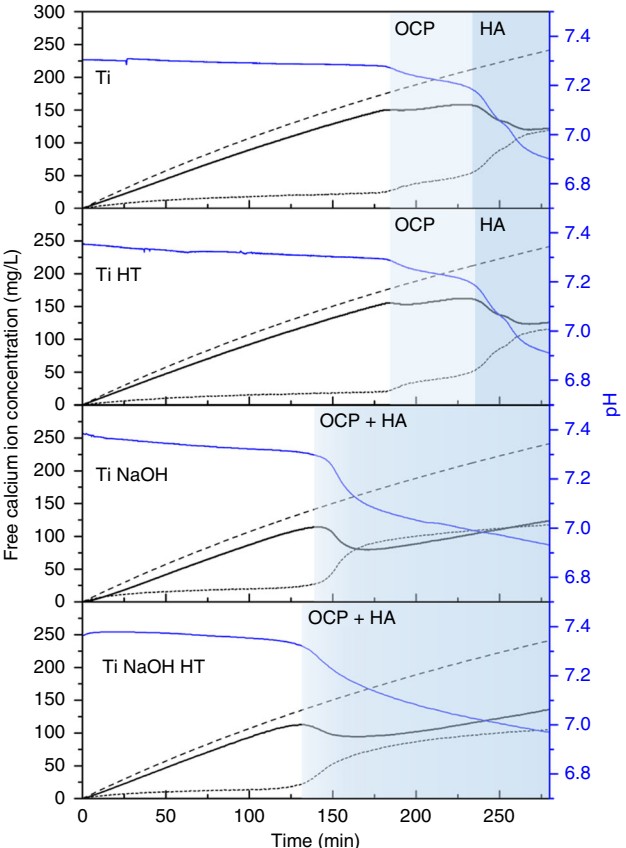

**Fig. 3** Free calcium profiles of titration experiments using four titanium surfaces. The black lines indicate free calcium concentrations measured by a calcium ISE. The dash lines give the concentration of added amount of calcium ions. The difference of the two is given by the dotted line. The colored region marks the possible dominant crystalline phases identified from TEM analysis

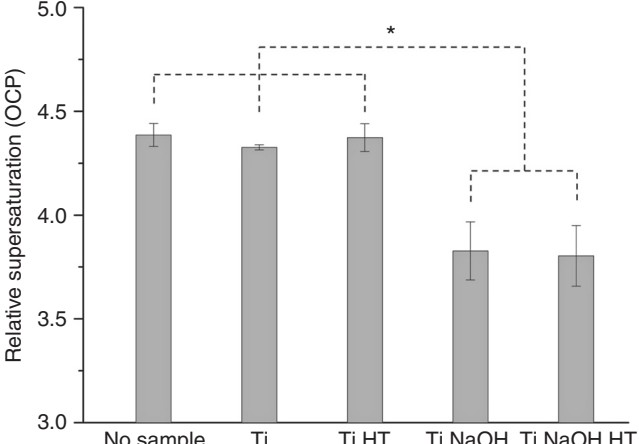

**Fig. 4** Solution relative supersaturation toward OCP at the first nucleation onset in different titration experiments. The relative supersaturation σ is calculated using the ratio of the activity product of ion units composing the crystal to the corresponding solubility product $K_{sp}$ considering the number of ion units (details in Supplementary Note 1[36]). Adsorption of free Ca ion by Ti NaOH and Ti NaOH HT is taken into account in the calculation, which slightly lowers the Ca concentration. pH changes due to the addition of powders are found to be negligible (<0.01) and thus not taken into account. Statistically significant differences between groups are marked with a (*) with $p$ value smaller than 0.01. The error bars are calculated standard deviations from three replicates in each case

not only the free calcium profile was modified with only one prominent peak remaining but also the nucleation onset point was significantly shifted to earlier time points. In addition, instead of a rather rapid nucleation process, the nucleation seems to happen slowly with a gradual increase in complexed and precipitated Ca around the onset point and a smoother pH profile. In fact, when sampled roughly 5–10 min before the highest free Ca point, crystalline CaP species can already be found together with pieces of Ti NaOH (Supplementary Fig. 6). At the highest free Ca point, crystalline CaP is also found embedded in the porous surface layer (Supplementary Fig. 7). The detailed analysis of the diffraction pattern indicates very similar features as the crystalline product from the homogenous nucleation, which is possibly OCP. At this stage, transformation to the thermodynamically most stable phase takes place and typical HA morphologies can be observed after the peak in free Ca even before reaching the lowest Ca concentration (Supplementary Fig. 8). Therefore, we propose that for the active surfaces of Ti NaOH and Ti NaOH HT, the initial nucleation barrier is significantly lowered and the free Ca peak observed corresponds to the final transformation, equivalent to the second peak in the case of homogeneous nucleation and nucleation in the presence of inert Ti and Ti HT. These features represent a clear measurable difference between different surfaces in solution. If we define the supersaturation of the solution at the first major precipitation event as an indicator for bioactivity, using the titration data and the thermodynamic model (Supplementary Note 1), we can quantitatively calculate the solution

supersaturation at the first nucleation point, taking into account the adsorption of free Ca²⁺ ion by Ti NaOH and Ti NaOH HT. Figure 4 shows a summary of supersaturation, calculated as for OCP, at the first free calcium peak for the four surfaces compared with the case of homogeneous nucleation ($n = 3$ using distinct samples). Statistical analysis using a one-way ANOVA test shows statistical differences between the five groups ($p < 0.01$). In addition, statistical significance exists between any case of the group of no sample, Ti and Ti HT and any material of the group composed of Ti NaOH and Ti NaOH HT ($p < 0.001$). There is no statistical difference between materials within each group. This conclusion is consistent with the direct visual observation of the modification of the free calcium profile from different materials but the supersaturation values provide us a quantitative measure of bioactivity. The lower relative supersaturation required for the precipitation in the case of Ti NaOH and Ti NaOH HT powders indicates their higher bioactivity compared with Ti and Ti HT powder. It should also be noted that this is in fact an underestimation for the bioactivity for Ti NaOH and Ti NaOH HT since the precipitation event actually corresponds to the second precipitation in the case of Ti and Ti HT. If in all cases we use the supersaturation level for the last precipitation event, which is the transformation to the thermodynamically most stable phase HA, the difference observed in the current experiments will be even more prominent.

**Comparison with static SBF test.** To compare our results to the traditional SBF test, regular static SBF tests were carried out with the same four Ti powders used in the titration experiment. Incubation was conducted at 3 and 7 days with three repetitions of each powder. The statistical design is presented in Table 1.

The precipitated amount of calcium onto the powder surface can be quantified by measuring the free calcium concentration of the remaining SBF solution using the Ca ISE. The results were analyzed using a multifactorial statistical model and presented in

**Table 1 Statistical design of experiments testing four titanium surfaces with SBF in static conditions**

| Factor | Definition | Level 1 | Level 2 | Level 3 |
|---|---|---|---|---|
| A | NaOH treatment | No | Yes | |
| B | Heat treatment | No | Yes | |
| C | Duration (day) | 3 | 7 | |
| D | Replicate | 1 | 2 | 3 |

**Table 2 Statistical design of experiments used in both torque tests as well as histology study. In each study, 48 implants are used**

| Factor | Definition | Level 1 | Level 2 | Level 3 |
|---|---|---|---|---|
| A | NaOH treatment | No | Yes | |
| B | Heat treatment | No | Yes | |
| C | Time delay (w) | 2 | 8 | |
| D | Position | Left | Right | |
| E | Replicate | 1 | 2 | 3 |

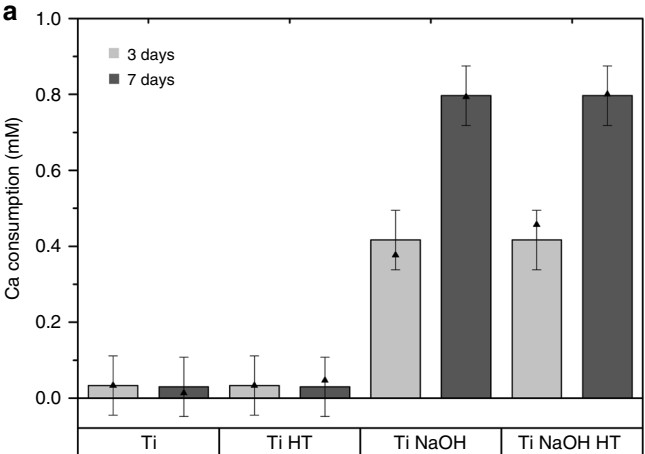

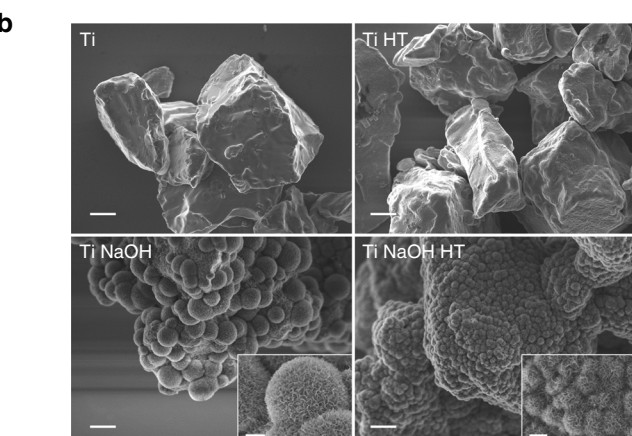

**Fig. 5** Apatite formation on different Ti surfaces. **a** The amount of calcium depletion at 3 and 7 days for different powders. The free calcium content is measured with Ca ISE and normalized with standard SBF solution with known concentration to obtain the depleted Ca. The column chart is obtained based on the fitted values using statistical analysis on the original data of three repetitions using distinct samples adopting a type I error risk of 5%. A 95% confidence interval is also given. The solid triangles give the experimental mean values based on three replicates. NaOH treatment and time duration are found to be statistically significant factors but not the heat treatment. **b** SEM images of Ti, Ti HT, Ti NaOH, and Ti NaOH HT after immersion in SBF for 7 days. Apatite formation can be identified by the presence of globules consisting of tiny flake-like crystals on Ti NaOH and Ti NaOH HT powder surfaces. Morphologies suggest a smaller globule size for the Ti NaOH HT powder compared with Ti NaOH powder. Scale bars are 5 μm in main images and 1 μm in the insets

Fig. 5a. No statistically significant changes in Ca concentration is observed in the case of Ti and Ti HT. On the contrary, Ti NaOH and Ti NaOH HT powders induced CaP formation as a function of incubation time since a decrease in solution calcium

concentration was observed. However, no statistical differences were found between Ti NaOH and Ti NaOH HT. SEM observations of the powders after incubation support the previous findings with NaOH-treated Ti powders (with or without heat treatment) fully covered with HA crystals, shown in Fig. 5b. These results are in good agreement with our findings using the titration method which successfully distinguished bioactive NaOH-treated Ti powders versus inert Ti and Ti HT powders.

**Correlation with animal experiments**. Animal experiments were conducted following a multifactorial experimental design, given in Table 2. This design was used for both torque test (48 implants) and histological studies (48 implants). A total of 96 implants were implanted into eight sheep. Full details of implantation site are given in Supplementary Table 2, 3 and raw data are given in Supplementary Table 5, 6. Except for the different chemical treatments and time delay before sacrifice, implants can also be grouped according to the implantation site, that is, either left or right side of the sheep pelvis. Three full replicates were used. When expecting no difference of the implantation site, the number of replicates becomes 6 for each implant type. The torque value of each implant measured individually gives a holistic indication of the implant integration into the bone environment. After statistical analysis of the data, among all the factors listed in Table 2, NaOH treatment ($p < 0.01$), time delay ($p < 0.01$) and the interaction of the two factors ($p < 0.05$) were identified to be statistically significant. Both heat treatment and implantation position did not affect the final torque value. Similarly, no statistically significant differences were found between different replicate groups, as expected. Combining these two factors, an overview of the torque value is given in Fig. 6a. Columns are the fitted values from the model and the solid triangles represent the experimental mean values based on six replicates. NaOH treatment is found to positively enhance the implant osseointegration, particularly for the 8-week group. Also, a longer healing time resulted in better mechanical property, as one would expect. This result is in excellent agreement with the previously discussed in vitro results which identified alkaline treatment and time delay to be the critical factor in conferring bioactivity to the Ti surface but not heat treatment. Similarly, the extracted stiffness values are also analyzed in the same way (Supplementary Fig. 9). In this case, only time delay ($p < 0.01$) was identified to affect the stiffness of the implant. A longer healing time led to higher stiffness, irrespective of the implant type. The analysis of the energy required to reach the yield point reached the same conclusion as the analysis of the yield, identifying NaOH treatment ($p < 0.01$), time delay ($p < 0.01$), and their interaction ($p < 0.05$) to be significant factors that systematically increase the energy (Supplementary Fig. 10).

The histological analysis of the implant was performed by the analysis of the bone-implant-contact (BIC) which reflects the osseointegration of the implants and is directly responsible for their biomechanical properties. The staining of the bone tissue

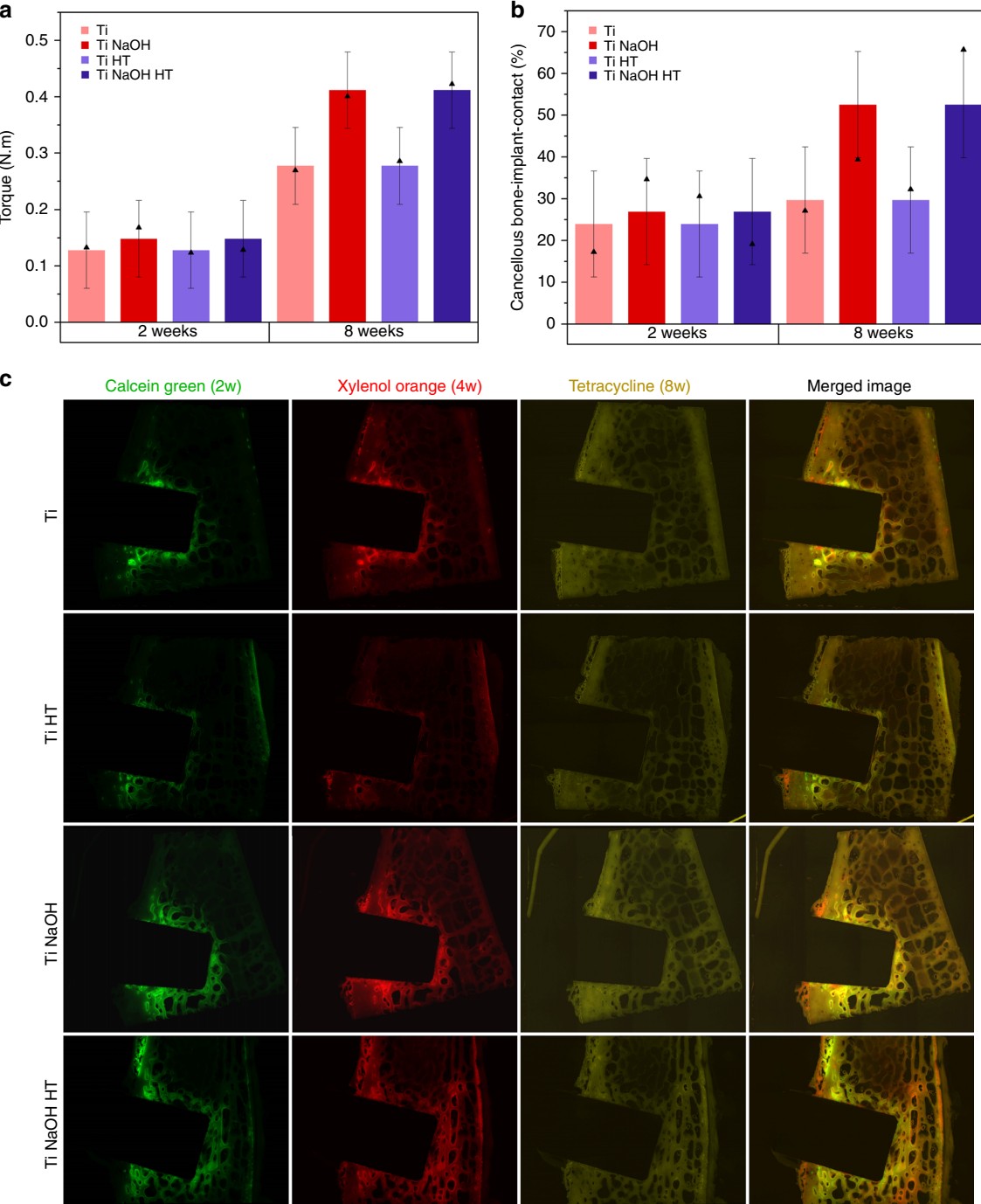

**Fig. 6** In vivo response for different Ti surfaces. **a** Control chart for the yield torque values and **b** Control chart for the cancellous bone-implant-contact. The columns give the fitted value according to a multifactorial statistical analysis with error bars corresponding to the 95% confidence interval. The solid triangles give the experimental mean values based on six replicates. For both tests, factor NaOH treatment ($p < 0.01$), time delay ($p < 0.01$), and the interaction of the two factors ($p < 0.05$) are identified to be statistically significant. Other interactions are treated as noise due to insignificant effect of factor heat treatment, position, and replicate. **c** Fluorochrome labeling of titanium implants after 2, 4, and 8 weeks for the 8-week group. The visible green regions represent the bone deposition 2 weeks after implantation. Similarly, the red regions represent 4 weeks deposition and yellow region represents bone deposition 48–72 h before the sacrifice (roughly 8 weeks after implantation). Merged images are provided in the right column. Since six replicates are used for each implant type, the images chosen are not necessarily a quantitative average representative of the bone remodeling process for each implant type. However, the same sheep (#04) is chosen in this case (based on image quality) to lower the inter-animal variability for enhanced consistency

provides a direct visualization of the bone tissue around implant and a quantitative measure of the BIC can be derived. The insertion of the Ti implant went through the cortical bone into the cancellous bone region, thus the BIC value can be separately determined for both cortical bone and cancellous bone. Figure 6b shows the summary of the analyzed BIC of the cancellous bone

tissue. Similarly to the findings from the torque evaluation, NaOH treatment ($p < 0.01$), time delay ($p < 0.01$), and interaction of both factors ($p < 0.05$) are identified to be significant. In addition, the interaction between heat treatment and time delay and a second order interaction between NaOH treatment, heat treatment, and time delay were also identified. However, since

heat treatment itself is not a significant factor, these interactions are treated as noise in the data presentation. The case of cortical bone provides a different result (Supplementary Fig. 11) showing only significant contribution from NaOH treatment ($p < 0.05$). The analysis of total BIC without distinguishing cancellous bone and cortical bone (Supplementary Fig. 12) produced the same conclusion as in the case of cancellous BIC. This is partly expected as the implant is mostly in contact with the cancellous bone and the corresponding BIC value would dominate in the overall BIC.

Fluorescent labeling of calcium provides a direct visualization of the bone remodeling process at different time points. A clear contrast between old and newly formed bone matrix can be obtained from fluorescent dyes. Figure 6c provides an example of the four Ti surfaces implanted for 8 weeks. It can be seen from the images that 2 weeks after implantation, bone remodeling was most intense at the peri-implant interface, especially the region of cancellous bone. After 4 weeks, although bone depositions near the implant–bone interface is still strong in some cases, more visible bone deposition occurred all over the bone tissue including both cancellous bone and cortical bone, a result of resorption and substitution. At 8 weeks, fluorescent oxytetracycline revealed a homogeneous distribution of newly formed bone in both cortical bone as well as in cancellous bone, indicating little preference for peri-implant bone remodeling. For the 2-week group, fluorescent labeling 48–72 h before sacrifice (roughly 2 weeks after implantation) revealed similar trend of intense peri-implant bone remodeling (The images for the 2-week group are given Supplementary Fig. 13).

**In vitro osteoblast responses.** Additional experiments using osteoblasts were conducted to investigate the interaction with osteoblasts using the four previously established surfaces in the form of discs plus one NaOH-treated Ti disc covered with HA fabricated by immersing in SBF for 5 days to obtain the HA coating. Bone morphogenetic protein 2 (BMP-2) was also introduced as an extra factor (experimental design given in Supplementary Table 4). In general, BMP-2 does not have a big influence on the cell viability. The number of cells continue to grow as a function of time. Despite some differences in the early stage such as 3 days, the number of cells are rather similar at 14 days. On the contrary, differences in alkaline phosphatase (ALP) activity, an enzyme produced by osteoblast and involved in the bone mineralization process, are more significant. In the presence of BMP-2, which is known to potently induce osteoblast differentiation, ALP activities are significantly increased compared with the case without BMP-2, especially at the 14-day point. However, at 14 days of incubation, the difference caused by surface chemistry seems to be masked by the effect of BMP-2 with no statistically significant difference observed between different surfaces. However, at 3 and 7 days, HA covered Ti NaOH disc stands out in ALP activity, showing that the surface coverage by HA layer has a positive influence on osteoblast activity in this case. Interestingly, in the absence of BMP-2 at 14 days, NaOH-treated Ti surface significantly promoted ALP activity compared with all other surfaces (Supplementary Fig. 14). Due to many possible conditions which can be used in the cell culture tests, the current results provide a snapshot of the potential of different surfaces to interact with osteoblasts and showed a generally positive effect of NaOH treatment and HA coating.

## Discussion

Although the titration using 20 mM calcium solution at a rate near 0.1 mL/min could seem to be slow, it should be noted that the nucleation event observed in the current experiments is a

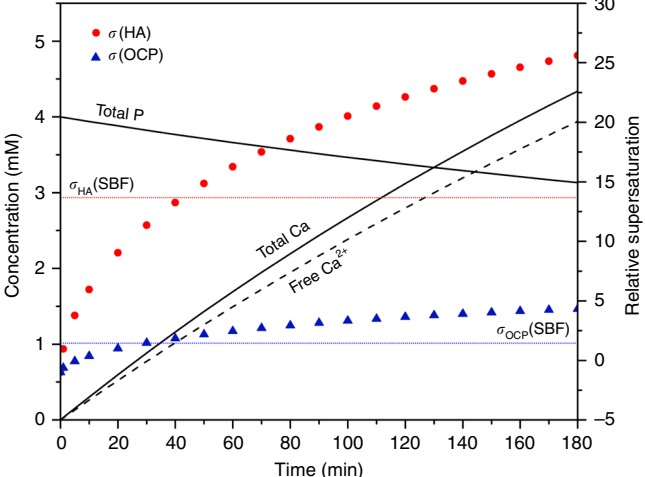

**Fig. 7** Theoretical calculation of solution supersaturations. The evolution of total Ca, free $Ca^{2+}$, total P concentration, and relative supersaturation $\sigma_{HA}$ and $\sigma_{OCP}$ as a function of time is calculated assuming no solid precipitates during titration. The difference between total Ca and free $Ca^{2+}$ is predominately attributed to the complexation in the form of $[CaHPO_4]^0$. Both HA and OCP become supersaturated within 6 min of titration. $\sigma_{HA}$ and $\sigma_{OCP}$ of standard SBF at testing conditions are given in two horizontal dotted lines

kinetically limited process. Due to a particularly low-solubility limit of HA, by the end of the first minute during titration, the solution is already supersaturated toward HA. In practice, the solubility limit is greatly increased for smaller HA particles due to curvature effects. Generally, the high energy barrier for HA nucleation helps to maintain stability in many metastable solutions including the standardized SBF, which is able to stay stable without precipitation of HA in detectable quantities for at least a month[23,24]. If we compare with the standard SBF, we find that after 40–45 min of titration the solution has surpassed the supersaturation level in SBF ($\sigma_{HA} = 13.62$), shown in Fig. 7. Although the two systems are rather different with different concentrations of Tris-HCl and particularly different Ca to P ratios (a starting phosphate concentration of 4 mM for titration vs. 1 mM in SBF), this time point can still be a general reference point and contrasts greatly with the very high level of supersaturation at the major nucleation events in homogeneous nucleation and in the cases where introduced materials have little effect on neither the pathway nor the rate of CaP nucleation. In fact, before reaching ~180 min where the first nucleation event is observed, if the titration is stopped, eventually the solution will precipitate CaP. By mixing 50 mL P solution with 5 mL Ca solution (corresponding to 65 min in the titration curve), short-term stability is achieved but an obvious precipitation event occurred after 10 h of mixing, as observed in the free calcium profile (Supplementary Fig. 15). Therefore, a consistent titration rate has to be used when comparing different materials. Some practical considerations for the use of the titration method is given in Supplementary Discussion. A titration experiment using commercially available hydroxyapatite particles is also presented for comparison (Supplementary Fig. 20).

The definition of bioactivity has gradually evolved over the years and is no longer limited to bone-bonding materials that were first discovered in the 1970s but can also refer to various new functional biomaterials[25,26]. In this work, we limit the discussion of bioactivity to the bone-bonding potential. The idea of using SBF to test implant materials is based on the observation that most bioglass that resulted in direct bone bonding typically

induced hydroxyapatite precipitation in vivo. In addition, it assumes that this phenomenon can be reproduced in vitro with carefully designed solutions. Therefore, under the in vitro context, the concept of bioactivity becomes an apatite forming ability. This idea of mimicking the body fluid might seem to be a natural approach at first. However, practically, it would be too ideal that a simple solution with mostly inorganic components is capable of reproducing implant behavior in the real body fluid, especially considering the significant effect of proteins. A strong inhibitory effect of proteins on HA formation has been reported with a concentration of 5 g/L, a concentration far below that in human blood plasma with the total protein concentration determined to be near 70 g/L[13,27]. Indeed, cases of inconsistency between in vitro predictions using SBFs and in vivo results have been reported[28].

In this work, our goal is not to mimic the body condition, but rather to measure the material's ability to promote HA nucleation with a quantitative output. The titration experiment in the time scale of a few hours is at least an order of magnitude faster compared with the traditional SBF test with a duration of 1–4 weeks. In fact, the experiment involving constant composition of solution, including the SBF test, where the induction time of HA formation is tested, as well as the constant composition approach developed by Nancollas, where the crystallization kinetics are studied at a fixed solution composition[29], can all be considered to some extent as a titration experiment with infinitely slow titration rate, except that only one solution concentration is used. On the other hand, our titration experiment adopts a general approach in testing the material where the material's ability to initiate CaP crystal formation is tested in varying solution compositions. The essential difference between an inert material and a bioactive material is their ability to trigger heterogeneous nucleation of CaP crystals on their surface. The substrate–solution and substrate–nucleus interfacial energy provide an overall energy advantage in the system for CaP crystal formation, either directly from the ions in solution, or from the attachment of amorphous species and then through amorphous–crystalline transformation, which is also highly relevant in the discussion of bioactivity due to the presence of amorphous CaP particles in natural biological systems and their function in biomineralization[30]. In fact, the possibility of testing amorphous CaP–implant interactions in the dynamic, and highly supersaturated titration system also opens up new opportunities for implant surface engineering, which, on the contrary, cannot be evaluated with traditional constant composition SBF. The varying solution composition in the titration system broadens the concept of in vitro bioactivity, making the evaluation no longer limited to the ability to induce heterogeneous nucleation from a specific artificial solution.

A very good correlation between in vivo and in vitro results using two separate methods is obtained in the current study, especially when comparing with total BIC and yield torque values obtained from the sheep model, which are the most relevant parameters since they provide a direct indication of successful osseointegration. All results highlighted the key factor of alkaline treatment in promoting the bioactivity of titanium implants. As naturally expected, longer implantation period also results in an increased implant stability, both biomechanically and histologically. However, another critical factor, that is the roughness of the implant, should not be neglected in the discussion of in vivo results especially the biomechanical performances. As can be seen previously, a porous surface in the micrometer-scale is obtained after the strong alkaline treatment of titanium surface, which might contribute to their better mechanical fixation. However, all the titanium bars received from commercial sources are in fact rather rough with features in the scale of a few microns.

The chemical treatment by NaOH thus does not significantly increase the roughness of the surface due to small porous morphologies masked by large existing features of the implant surface (3D surface profiles and quantitative analysis of surface roughness are given in Supplementary Fig. 16, 17). In addition, the consistent results of BIC both in the cancellous bone and cortical bone with the torque value indeed shows that the better mechanical fixation of the implants that underwent NaOH treatment or were implanted for a longer period of time is indeed very likely to be a result of larger area of direct contact with the surrounding bone that was a result of intense bone remodeling process, rather than from any minor differences in roughness.

The current method also serves as a platform to investigate the effect of additional components such as proteins and amino acids. With tunable factors such as the titration rate, initial phosphate concentration, calcium solution concentration, and pH buffering mechanism, potential optimization of the method can be expected. It should also be particularly noted that in the current study using four Ti surfaces, the same conclusions were obtained using animal experiments, traditional SBF and the titration method but with significantly different time duration. In the case of animal experiments, at a sacrifice of eight sheep, implants were evaluated after 2–8 weeks of implantation. In the case of static SBF test, implant surfaces were studied after 3–7 days of immersion and the general recommendation in the ISO protocol is 28 days[8]. In the titration experiments, only 5 h are needed for inert surfaces and active surfaces can be studied within 3 h, meanwhile producing a quantitative measure of their bioactivity. This is a significant step forward for the community in saving time and costs while expanding the breadth of materials that can be tested in a quantitative manner.

In summary, the current study compared the behaviors of different titanium powders in a calcium titration setup and observed that neither Ti nor heat-treated Ti powder could induce statistically significant modifications in calcium phosphate nucleation compared with a homogeneous nucleation system. However, the introduction of NaOH-treated titanium or NaOH plus heat-treated titanium powder significantly modified the nucleation barrier and facilitated the nucleation event. The measured bioactivity, as quantified by the relative supersaturation at the nucleation onset, is determined to be statistically higher than that of Ti and heat-treated Ti powder. NaOH treatment was identified as the key process to induce bioactivity for Ti powders. A standard static SBF test produced similar results, showing apatite formation on Ti NaOH and Ti NaOH HT powder after 3 days of incubation, whereas Ti and heat-treated Ti powder failed to produce HA even after 7 days. Both tests did not show any effect from heat treatment in terms of modifying the bioactivity.

Using a pelvic sheep model, implanted titanium bars with four different surface treatments were evaluated by biomechanical tests and histological studies at 2 and 8 weeks implantation period. The yield torque values and the energy to yield point of all implants studied reached similar conclusions as the in vitro studies that both NaOH treatment, time and their interaction are statistically significant factors in promoting a better osseointegration of the implant. The stiffness of the implant is, however, only positively correlated to the implantation time. NaOH treatment and time of implantation are also significant factors for the bone-implant-contact in the cancellous bone, while for cortical bone, the only significant effect identified was from NaOH treatment. The analysis of total bone-implant-contact produced the same conclusion as that of the cancellous bone. The consistent conclusions from both torque test and histology studies suggest that the higher bone-implant contact (BIC) values are directly responsible for the enhanced mechanical properties.

In addition, the bone remodeling process was revealed by fluorescent labeling in which significant new bone regeneration at the peri-implant interface within 2 weeks of implantation was observed. The proposed titration method was further corroborated with cell culture results using osteoblasts, which showed significant promotion of alkaline phosphatase (ALP) activity by NaOH-treated Ti disc at 14 days when no bone morphogenetic protein 2 (BMP-2) is added. A significant ALP activity was also observed for NaOH-treated Ti disc covered with HA compared with other surfaces at 3 and 7 day time points when 32 nM of BMP-2 is introduced.

The rapid and quantitative titration method proposed in this work provides a different approach for the evaluation of bioactivity of implant materials. Potential improvements of the protocols and optimization of testing parameters are currently being investigated.

## Methods

**Materials preparation**. Four titanium-based surfaces were chosen for this study: (i) titanium with no chemical treatment, (ii) titanium—heat-treated in air, (iii) titanium—treated with NaOH solution, and (iv) titanium—NaOH treated and heated in air. For the titration and static SBF experiments, commercial Ti powders (−325 mesh, 99.5%, ABCR, Germany) are used. NaOH treatment was done with pre-heated 5 M NaOH solution for 2 h at 60 °C (1.0 g of powder in 20 mL NaOH solution). Afterward, the powder was washed with de-ionized water for three times for ~30 s each and one time with ethanol for ~30 s and then dried in 60 °C oven. The heat treatment was applied at 600 °C (heating rate 100 °C/h, natural cooling) for an hour in ambient atmosphere. For sheep experiments, commercial pure (CP) grade 2 titanium bars (Hempel Special Metals, Switzerland) of ø 6 mm × 10 mm were first washed with acetone, water, and ethanol in an ultrasonic bath before chemical treatments. No polishing was applied on the surface. The same four treatments as for the in vitro experiments were used. The NaOH treatment was conducted using 5 M NaOH solution for 24 h at 60 °C followed by washing for three times ~30 s each with flowing water and drying under ambient conditions. The heat treatment was applied at 600 °C (heating rate 100 °C/h, natural cooling) for an hour in ambient atmosphere. Biomechanical torque tests and histological studies were carried out using separate implants. For the implants used for torque tests, the head of the implant was specifically fabricated into a hexagonal shape for mechanical fixation with the rotary actuator during torque tests. This hexagonal part has a length of 3 mm and was capped with a PEEK healing cap after implantation. The other half of the implants for histology were used directly in the shape of bars. In all cases, implants were inserted into the bone for 7 mm (Supplementary Fig. 18). Before implantation, all implants were plasma-sterilized at the Vetsuisse Faculty Zürich, Switzerland. Materials preparation for cell culture experiments is described in detail in Supplementary Methods.

**Titration experiment**. Detailed experimental protocols of the titration method are given in the Supplementary Methods. In general, calcium stock solutions (containing 100 mmol NaCl, 20 mmol CaCl₂, 20 mmol Tris, and 17.5 mL of 1.0 M HCl solution per liter of solution) are titrated dropwise into 50 mL phosphate solution (containing 100 mmol NaCl, 4 mmol Na₂HPO₄·2H₂O, 20 mmol Tris, and 17.5 mL of 1.0 M HCl solution per liter of solution) under constant magnetic stirring. The titration rate is kept constant for different materials near 0.1 mL/min. The real titration rate in all experiments was calibrated to be 0.077 mL/min. During the titration experiment, the temperature is controlled at 25.0 ± 1.0 °C. The starting pH of the solution is calculated to be 7.45 using a thermodynamic model (Supplementary Note 1). The pH, free calcium ion concentration, and temperature are monitored in situ and recorded by a combined pH electrode with a temperature probe (InLab Expert Pro, Mettler Toledo) and calcium ion selective electrode (perfectION Ca combination electrode, Mettler Toledo). Four titanium powders were tested in the titration experiment. In total, 0.1 g of powder was used for 50 mL starting volume of phosphate solution.

**SBF test**. Four titanium powders were tested with standard SBFs, prepared by strictly following the procedures of ISO 23317:2014 and contains 142.0 mM Na⁺, 5.0 mM K⁺, 1.5 mM Mg²⁺, 2.5 mM Ca²⁺, 147.8 mM Cl⁻, 4.2 mM HCO₃⁻, 1.0 mM HPO₄²⁻, 0.5 mM SO₄²⁻, and buffered by adding 50 mmol Tris and 39 mL of 1.0 M HCl to make one liter of solution[8]. The standard SBF test was conducted by mixing 20.0 mg of powder with 30 mL of SBF followed by incubation at 37.0 °C for a duration of 3 or 7 days. The amount of apatite precipitated was quantified by analyzing the calcium concentrations of the remaining SBFs after the test using a calcium selective electrode (perfectION Ca combination electrode, Mettler Toledo). Due to the large size of titanium powders, most powder sedimented after the incubation period. The supernatant was withdrawn and filtered through a 0.2 μm surfactant-free cellulose acetate syringe filter (Thermo Scientific) to eliminate any possible powder in the remaining liquid. Afterward, 0.050 mL 65 wt.% HNO₃ is

added to each remaining liquid of 20 mL to prevent possible CaP precipitation which would decrease the free calcium concentration after the incubation period. Finally, the solution is measured with calcium ion selective electrode (ISE) for its free calcium concentration. The low pH (< 2) from the addition of acid also releases Ca²⁺ from minor complexation with phosphate species resulting in more than 99.5% of Ca existing in the form of free Ca²⁺ as calculated by the thermodynamic model. The minor change in Ca complexation due to difference in phosphate concentration is negligible at such pH levels. Thus the free Ca²⁺ concentration of each solution is directly used without further correction for each solution as the total Ca remaining in the solution. As-prepared SBFs without the addition of any materials are introduced as control solutions and the amount of calcium precipitated is then calculated as the difference with the value obtained from the control group after normalization.

**Materials characterization**. X-ray diffraction (XRD) analysis was conducted using a Philips X'Pert X-ray diffractometer with Cu K radiation (Kα1, λ = 1.5406 Å) operating at 45 kV and 40 mA. Crystal structures of the tested materials were identified using standard profiles from the JCPDS database. Surface morphologies were observed by scanning electron microscopy (SEM, Merlin (GEMINI II), Zeiss). Powder samples from the titration were observed by transmission electron microscopy (TEM, FEI Tecnai Osiris) operating at 200 kV.

**Experimental animals and surgical model**. Eight adult, female Swiss alpine sheep were used for this study (average age: 3 years, average weight: 79.4 kg). The animals were split into two groups, one group was sacrificed after 2 weeks (n = 4) and the other after 8 weeks (n = 4). Experiments were conducted according to Swiss laws of animal protection and welfare and approved by the local federal veterinary council (authorization: ZH108/16). Implantation was conducted at the iliac wing of the sheep using a well-established and previously reported surgical method[31–34]. The implants were placed in the cranial part of the left (n = 6) and right (n = 6) iliac wing of all 8 sheep, whereas implants in each iliac wing were only used either for biomechanical tests or histological studies. This design allows 12 implants in each sheep and a total of 96 implants. In each surface group (Ti, Ti HT, Ti NaOH, Ti NaOH HT) with 2 or 8 weeks implantation, six implants were examined histologically for their bone-implant-contact (BIC) and six implants were tested biomechanically using a torque test model. All details regarding the surgical procedure including anesthesia, surgical operation, postoperative care and harvesting of the implants are given in Supplementary Methods.

**Torque test and data analysis**. The cut specimens were embedded in acrylic glass (SCS-Beracryl D-28, Suter Kunststoffe AG, Switzerland) in 5 × 5 × 2 cm square forms. When hardened, these specimens were placed in an Instron E10000 test machine (Instron, USA). The axis of the machine head was aligned with that of the implant before the hexagonal implant head was locked into the Torsion-Measurement-Unit (Model D-2209, Lorenz Messtechnik, Germany). Removal torque testing was performed with a speed of 0.1°/s until failure. The torque was recorded in N·m to a maximum rotation of 30°. All data were analyzed using a MATLAB code to find three values: (1) Yield point [N·m] (border of elasticity): the point of irreversible deformation and failure of the bone-implant-contact defined as the torque value at which the slope of the curve is 50% of that of the linear region of the curve. (2) Rotational stiffness [N·m/°]: the slope of the linear region of the curve which reflects the stiffness of the implant–bone connection against elastic deformation. (3) Energy to yield [J]: The energy that the implant–bone connection was able to absorb until the border of elasticity calculated using the area under the curve (example using Ti NaOH HT implant (8 weeks) given in Supplementary Fig. 19).

**Histological preparation**. Specimens for histology were fixated in 40% ethanol and further dehydrated in solutions with ascending ethanol concentration (50, 70, 80, 90, 96, and 100%). During the following 7 days they were kept in xylene under vacuum before being soaked and then embedded in poly(methyl methacrylate) (PMMA). The blocks were cut with a diamond saw so that multiple ground sections of the longitudinal axis of the implant could be obtained. Native sections were glued onto standard glass slides while sections for staining were glued onto plexiglas slides and polished to 100–110 μm thick and then stained with toluidine blue. These stained ground sections were photographed and digitalized using a macroscope (Macroscope Z6 APO A, Leica Microsystems, Switzerland) and a mounted camera system (Leica DFC 450 Digital Camera, Leica Microsystems, Switzerland).

**Bone-to-implant contact (BIC) determination**. All histological slices were digitalized with a magnification of 1.0 × 10.9 and evaluated individually. The entire implant surface implanted into the bone was measured using a specialized software (Image IMS, Imagic Bildverarbeitungs AG, Switzerland). The lines of direct bone-to-implant contact were marked and its percentage in comparison with the total implant surface was calculated for every implant. Both the BIC in the cortex and cancellous bone were measured separately and total BIC was also calculated.

**Observation of bone remodeling using fluorochrome labeling**. Fluorescent dyes that can bind irreversibly to calcium in the bloodstream were used to follow the dynamic bone remodeling process by identifying the hydroxyapatite deposition over time. The fluorescence labeling was performed by subcutaneous injection using calcein green at 5 mg/kg body weight (BW), xylenol orange at 90 mg/kg BW and oxytetracycline (Engemycin, MSD Animal Health, Switzerland) at 20 mg/kg BW. For the 2-week group, only calcein green was given 48–72 h before sacrifice. For the 8-week group, calcein green was given after 2 weeks of implantation, xylenol orange was given after 4 weeks of implantation and oxytetracycline was given 48–72 h before sacrifice, roughly 8 weeks after implantation. Fluorescent dyes were later detected in ground histology sections with a fluorescence microscope (LeicaDM6000B, Camera DFC350 FX, Leica Microsystems, Switzerland).

**Statistical analysis**. Multifactorial statistical design and analysis of experiment were used throughout the text, statistically significant differences were identified using a full factorial statistical model[35]. The model uses least squares regression methods to analyze the main effects as well as the interactions between factors adopting a type I error risk of 5%. The significance of regression coefficients was tested by analysis of variance (ANOVA). A linear regression is applied to two-level factors and their interactions; quadratic regression is applied to three-level factors and their interactions with other factors. This factorial statistical model was also used in the sheep experiments which allows the use of minimal number of animals to identify statistically significant factors. In the analysis of relative supersaturation and cell culture experiments, statistically significant differences between different groups were measured using one-way analysis of variance (ANOVA) followed by Tukey's multiple comparisons. A $p$ value smaller than 0.01 is considered statistically significant.

**Reporting summary**. Further information on experimental design is available in the Nature Research Reporting Summary linked to this article.

## Data availability

The data that support the findings of this study are available from the corresponding author upon reasonable request.

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

## Acknowledgements

The authors would like to thank Swiss National Science Foundation (Project 205321 – 150193) for providing the financial support of this work.

## Author contributions

W.Z., P.B., and B.v.R. conceived and designed the study. W.Z. performed the titration experiments, titanium sample preparations, data analyses and wrote the manuscript. D.M. and B.v.R. carried out the animal experiments and contributed to the writing of the methods section. S.F. conducted the torque tests. W.H. carrier out the cell culture tests. J.L. assisted in data analysis. All authors discussed the results and contributed to the interpretation of data.

## Additional information

**Competing interests:** The authors declare no competing interests.

