## [Peer Review File · Nature Communications]

Reviewers' comments:

Reviewer #1 (Remarks to the Author):

Manuscript natcom_181192, "Rapid evaluation of bioactive Ti-based surfaces using an in-vitro titration method".

The paper deals with an improved protocol for bioactivity evaluation of Ti-based surfaces. The aim is clear and relevant. The topic is well introduced and the experimental study is addressed from different prospective, from titration to in-vitro to in-vivo tests. Moreover, the author's conclusions are supported by an accurate statistical analysis.

The method allows assessing bioactive or non-bioactive surfaces: an example for not active surface (Ti or Ti HT) and active surface (Ti NaOH or Ti NaOH HT) is provided. The outcomes of the tests themselves are actually not a novelty, since it was known that the two tested surfaces have such properties. The point is that the assessment can be done in few hours instead of some days, such as following the ISO23317.

Nevertheless, few statements seem not appropriate and few clarifications should be provided before publishing:

- 1) Ca solution flow rate: in the main text, section "titration experiment" (P4,L8), it is reported a value of 0.1 mL/min. In SI the reported value is 0.08 (P2,L5) and 0.077 (P9,L19). Flow rate is a fundamental parameter and even a limited change can have significant impact on the precipitation pathway. Please, specify the flow rate used. In case of differences, please specify why different values were applied;
- 2) A recent paper was published by Carino A. et al. [1] on calcium phosphate precipitation. This reference should be mentioned on P2,L20 (with refs 14-15) and on P22,L20 (with refs 27, 32);
- 3) Why do the authors carried out the measurements at 25 C instead at 37 C which might be more consistent with the mentioned ISO standard and more meaningful considering the aim of the test? Moreover, I guess the "in-vitro osteoblast response" (P20, L10) was carried out at 37 C;
- 4) P7L32: "... due to limited amount of phosphate..." as shown in the mentioned paper [1] at this stage the detected amount of free Ca²⁺ depends on the growth rate and the Ca²⁺ addition rate. If the addition rate is slow enough, the obtained value corresponds to the thermodynamic equilibrium (solubility) of the most soluble solid phase in the system. From this value the "true" saturation can be estimated. By liner extrapolation, it can be seen that the system under analysis is supersaturated after about 75 min (or 6 mL of Ca²⁺, fig 1), with respect to the solid phase present in the timeframe 300-350 min. It should be useful to plot the calculated S value, based on the TD model used, and the calculated and measured solubility values.
- 5) Solid before the first jump (180 min, figure 1). The authors claim the presence of solid particles before the first jump, and even before (at around 120 min). As the authors stated, these particles might be due to the drying process during (conventional) TEM samples preparation. Therefore, are clearly an artefact. I would recommend to do not mention the "formation of ACP" based on such findings because is misleading for non-experts and in contradiction with the reported experimental data for experts (for instance, the TD model do not consider ACP formation before the first jump, is that correct?). For sure, the fast drying of a phosphate buffer solution after, e.g., 15min of Ca solution addition, will promote the formation of ACP. Maybe cryoTEM can help here, but it has been done several times in many papers, with non-reproducible results. Therefore, I simply suggest to do not mention it; it does not add anything to main manuscript message;
- 6) OCP and HA. The authors claim a two phase precipitation process. This has been reported in literature in many papers. Figure 1 is a strong evidence of it. Figure 3a/b show, maybe, even a third phase. The multiple-stage is a matter of fact. Nevertheless, the nature of the solid phases in each stage is not so clear. In my opinion the experimental evidences are not sufficient to support the statements about which phase is present at each stage. The most intense diffraction peak which clearly discriminate between OCP and HA occurs at too small angle to be detected by SAED (masked by the beam stop). All the other peaks overlap and cannot be resolved by SAED.

Moreover, many papers claim that OCP and HA coexist in a broad range of experimental conditions. Figure 2 (and the related SAED), unfortunately, does not help much to corroborate the author's speculations (here, I would exclude figure 2e/f for the reason of point 5). The experimental evidences show two morphologies. That is quite clear. But, in my opinion, not more than that. The small ED differences might be a consequence of the morphology as well as any other non-stoichiometric compounds in the continuous transformation between OCP and HA. Interestingly, in the experimental conditions studied, no ACP is detected (excluding the artifact due to drying). A double steps precipitation is often considered: first ACP and then a crystalline phase, according to the Oswald step rule. But it is not the case here, as also reported in [1]. The formation of DCPD cannot be excluded too (with acicular morphology). The first step seems consistent with the formation of a rather soluble solid phase (stage 190-250) followed by a second solid with lower solubility (280-350). From these almost linear sections, the solubility product can be calculated and compared with the literature values, in order to try to identify which phase might be present at that stage. Nevertheless, the so-calculated solubility values refer to the most soluble solid phase (whereas the diffraction patterns depict a volume averaged composition). The point here is that it does not matter which phase is produced: the value of the manuscript, in my opinion, is that there is a measurable difference between the nucleation conditions with or without an active surface. Therefore, I suggest to smooth the statement about the phase obtained because the presented evidence (in this manuscript as well as in many other papers on this topic), does not allow to line out clear statements (unless the diffraction peak at low angle, unique of the OCP structure, can be shown);

7) SI, chapter 8. Actually, already Figure 3 strongly support the findings reported in this chapter. For instance, in Fig 3c, an offset in the linear part Ca free curve is shown. By linear (range 25-125 min) extrapolation to time=0, a negative Ca free concentration can be measured, which should correspond to the amount of adsorbed Ca on the Ti NaOH surface. Doing the same exercise on the other curves of the same figure, it is clear that a very small amount of Ca is adsorbed in the case Ti NaOH HT and no adsorption occurs in casa a and b. I suggest to compare the values obtained from Fig 3 and the values reported in table S4;

8) SI, table S4; typo: Mes 1 \diamond Mes 2;

9) SI, P9L29; Typo: Fig. S4 \diamond S6

10) SI, Figure S12: "indicating a gradual nucleation". This is conceptually incorrect. In this case, the process is just growth, not even heterogeneous nucleation; it is just continuous growth of HA on HA (as state on P23L11).

11) General consideration: Figure 1 vs. Figure 3. In Figure 1 as well as Figure 3a and 3b, homogeneous nucleation occurs. Therefore, Ti and Ti HT are "not active". In case of figure 3c and 3d, heterogeneous nucleation occurs instead. Therefore, a lower S is required to initiate the solid formation. Heterogeneous nucleation occurs because of an extra cohesion (or adhesion) energy contribution is provided during the nucleation stage on the Na-titanate layer. Thus, these surfaces are "active", i.e. they show an apatite-forming ability. The solid formation on-set, define the "activity": if it happens before the homogeneous threshold, the surface can be defined "active" since heterogeneous nucleation occurs. The fact that the precipitation is "less sharp" is a consequence of the different nucleation (and growth) rate, consequent of the lower saturation level (page 13 lines 5-9). The solid formed at lower S can be different; in fact, the equilibrium stage (e.g. 175-300 min, in fig, 3c/d) corresponds to a solid phase that have a solubility similar to the solid phase obtained at the end (275 min) of fig 3a/b. Therefore, in the presence of an active surface, the intermediated phase of fig 3a/b is not formed (in agreement with the author's statement, P13L17-19). Because of heterogeneous nucleation, of the most thermodynamic stable phase is promoted. In my opinion, the concept of heterogeneous nucleation should be better introduced and discussed in the paper, since is the appropriate term which justifies the results. Instead, the section on P24L2-15, in my opinion, is rather misleading since (i) the influence of ACP (i.e. the artefact in the context of this manuscript) as well as (ii) pre-nucleation clusters (which are out of the scope of this manuscript) are recalled and the (iii) heterogeneous nucleation concept (the key point of the manuscript) is considered not much relevant to assess the surface activity. In my opinion, the "material affinity to CaP" is "the ability to promoted heterogeneous nucleation";

12) I'm not able to evaluate the correctness of the in-vivo test. Nevertheless, the procedure and

the explanation of the results are clearly described and convincing.

[1] <https://doi.org/10.1016/j.actbio.2018.05.027>

In conclusion, hoping that the authors find my criticisms constructive, I enjoyed the manuscript and I think is worthy of publication. Therefore, I would suggest elaborating the comments, smooth some statements which are not fully supported by experimental evidences or not relevant to support the major message of the manuscript; I would be happy to see a revised version.

Best regards.

Reviewer #2 (Remarks to the Author):

It is an interesting and important topic to develop in vitro method to evaluate the in vivo performance of biomaterials. SBF method has been widely used because its ability to mimic the biological ion concentration and temperature, although it does have some limitations. In order to address these limitations, this study aimed to investigate a calcium titration method as a substitute for SBFs to evaluate the in vivo bioactivity. The paper was well written and organized, however the advantages of titration method was still debatable, compared to other in vitro methods such as some modified SBFs and the constant composition method developed by Dr. Nancollas. Some questions or comments are listed as follows.

- 1) The titration process could not mimic the biological condition that has relative stable ions strength and temperature. The different environments (such as Ca/P ratios, the presence of Mg^{2+} , and solution temperature...) may result in a different mineralization pathway for calcium phosphates. It might not be suitable to reflect the in vivo situation.
- 2) It is not clear whether this method is sensitive enough for other materials having moderate bioactivity, rather than the Ti-based materials which are well-studied bioactive materials.
- 3) One big challenge of SBF is the inhibitory mineralization in the presence of proteins. It is not clear if same effect would happen when protein is in this titration system.
- 4) The decrease of free Ca^{2+} was measured to follow the mineralization process, but how to ensure that the Ca^{2+} drop was caused by the mineralization of surface, rather than the mineralization in the solution?
- 5) It was noted in Fig.3 that the initial pH values were different in these groups. The groups of Ti with NaOH treatment have shown higher initial pH values. Would this higher pH affect the mineralization in the systems? Was the washing process (rinsed by DI water for 30s, 3 times) thorough enough to remove the NaOH residues from the sample surfaces?
- 6) The supersaturation degree towards OCP was calculated to indicate the bioactivity, however there is no direct evidence to prove the presence of OCP in Figure 3. Did the authors conduct the HR-TEM studies for the early formed crystals?

Reviewer #1 (Remarks to the Author):

Manuscript natcom_181192, “Rapid evaluation of bioactive Ti-based surfaces using an in-vitro titration method”.

The paper deals with an improved protocol for bioactivity evaluation of Ti-based surfaces. The aim is clear and relevant. The topic is well introduced and the experimental study is addressed from different prospective, from titration to in-vitro to in-vivo tests. Moreover, the author’s conclusions are supported by an accurate statistical analysis.

The method allows assessing bioactive or non-bioactive surfaces: an example for not active surface (Ti or Ti HT) and active surface (Ti NaOH or Ti NaOH HT) is provided. The outcomes of the tests themselves are actually not a novelty, since it was known that the two tested surfaces have such properties. The point is that the assessment can be done in few hours instead of some days, such as following the ISO23317.

Nevertheless, few statements seem not appropriate and few clarifications should be provided before publishing:

1) Ca solution flow rate: in the main text, section “titration experiment” (P4,L8), it is reported a value of 0.1 mL/min. In SI the reported value is 0.08 (P2,L5) and 0.077 (P9,L19). Flow rate is a fundamental parameter and even a limited change can have significant impact on the precipitation pathway. Please, specify the flow rate used. In case of differences, please specify why different values were applied;

The flow rate used was constant in all experiments at 0.077 mL/min. This has been corrected in the SI. In the main text, however we recommend to write: “The titration rate is kept constant for different materials near 0.1 mL/min. The real titration rate in all experiments was calibrated to be 0.077 mL/min.” We recommend to keep “near 0.1 mL/min” for simplicity and easier to remember, but the actual flow rate of 0.077 mL/min is also clearly specified right after.

2) A recent paper was published by Carino A. et al. [1] on calcium phosphate precipitation. This reference should be mentioned on P2,L20 (with refs 14-15) and on P22,L20 (with refs 27, 32);

This citation was added to the manuscript as requested.

3) Why do the authors carried out the measurements at 25 C instead at 37 C which might be more consistent with the mentioned ISO standard and more meaningful considering the aim of the test? Moreover, I guess the “in-vitro osteoblast response” (P20, L10) was carried out at 37 C;

We started off our tests at 25°C for simplicity since the aim of this titration method was not to mimic the body condition but to assess the material’s ability to promote the heterogeneous nucleation of crystalline calcium phosphates, an ability that is closely related to the *in vivo* bioactivity. When the idea is not to mimic the body condition, it is thus not critical to use 37°C. In fact, the complexity of the body condition makes the complete simulation not practically feasible, and adding the organic content and tuning other experimental parameters (such as phosphorus concentration, temperature etc.) represent the next step in exploring a better optimization of the current protocol.

Nevertheless, our preliminary experiments at 37°C indicate a similar onset point for homogeneous nucleation (no presence of sample) at around 180 min whereas the presence of Ti NaOH also greatly shifted the nucleation barrier towards earlier time points around 60 min, shown in the figure below. In this case we also see that the homogeneous two stage nucleation event has been reduced to single stage for homogeneous nucleation (only one peak left). In fact, the shape of the curve can also be a result of a two stage event which happened very close to each other. In general, this minor difference from our observation at 25°C is likely to be from a small difference in crystalline transformation kinetics. In fact, performing the test at 25°C can actually provide us a better chance to study this transformation in more detail and thus be able to distinguish the subtle differences in materials bioactivity. However, in order to systematically compare the two temperatures, much further work is needed. Nevertheless, the current testing protocol at 25°C has already given us a convincing distinction between bioactive and bio-inert materials which is also corroborated by the SBF test, *in vitro* cell culture tests and *in vivo* animal experiments, supporting our choice.

Fig. Development of the free calcium ions measured by the calcium ISE (black line) in comparison with the added amount of calcium ions (red line) for control group with no samples (a) and testing group with Ti NaOH (b). The difference of the two lines indicates complexed and precipitated calcium, shown in the blue dash line. Temperature is kept at 36.5 ± 1.0 °C. The preparation of the solutions follows the same procedures as the main text.

The cell culture tests were conducted at 37°C following the standard practice in cell culture studies. The missing details are now added to the relevant experimental section.

4) P7L32: "... due to limited amount of phosphate..." as shown in the mentioned paper [1] at this stage the detected amount of free Ca^{2+} depends on the growth rate and the Ca^{2+} addition rate. If the addition rate is slow enough, the obtained value corresponds to the thermodynamic equilibrium (solubility) of the most soluble solid phase in the system. From this value the "true" saturation can be estimated. By liner extrapolation, it can be seen that the system under analysis is supersaturated after about 75 min (or 6 mL of Ca^{2+} , fig 1), with respect to the solid phase present in the timeframe 300-350 min. It should be useful to plot the calculated S value, based on the TD model used, and the calculated and measured solubility values.

The solubility line of Ca can be calculated, assuming no P is consumed. Due to the dilution of the system thus a decreasing P concentration, the Ca solubility line goes up. This line serves as a reference for supersaturation since once the measured free Ca^{2+} is higher than the calculated value, the system is supersaturated toward a specific phase. Just like in the reference paper [1], the system quickly reaches supersaturation versus HA (solubility line in the bottom) and the phase of DCPD is more soluble. However, once CaP is precipitated, the solubility line is more difficult to predict, unless the free P concentration is also monitored in real-time and the measured solubility value of a certain CaP phase is shown to be constant over time. In our system, however, the growth of the existing solids are not yet terminated if we look at the dotted line of Fig. 1 indicating complexed and precipitated Ca which continues to increase despite the dilution of the system. We feel that the plateau is not sufficiently stable or long enough to have reached a steady state, which makes an accurate estimation challenging. Therefore, we think that giving the supersaturation level is an alternative way to describe the system. In fact, the thermodynamic model has been used to predict the supersaturation level for OCP and HA assuming no precipitates in Fig. 10. In this plot, we can also see the reference supersaturation level in Kokubo's SBF, directly comparing the two systems.

5) Solid before the first jump (180 min, figure 1). The authors claim the presence of solid particles before the first jump, and even before (at around 120 min). As the authors stated, these particles might be due to the drying process during (conventional) TEM samples preparation. Therefore, are clearly an artefact. I would recommend to do not mention the "formation of ACP" based on such findings because is misleading for non-experts and in contradiction with the reported experimental data for experts (for instance, the TD model do not consider ACP formation before the first jump, is that correct?). For sure, the fast drying of a phosphate buffer solution after, e.g., 15min of Ca solution addition, will promote the formation of ACP. Maybe cryoTEM can help here, but it has been done several times in many papers, with non-reproducible results. Therefore, I simply suggest to do not mention it; it does not add anything to main manuscript message;

Indeed, the presence of clusters in Tris-buffered supersaturated Ca-P solution system is currently under debate. With the reference in literature published by N. Sommerdijk in 2010 in Nat. Mater. using cryo-TEM [2] and early evidences from other group using enhanced-DLS [3], we tend to believe the presence of clusters in the system. However, their quantity and how much they influence the crystal nucleation remains unclear. And the drying process in conventional TEM preparation can certainly change their native state which result in the amorphous beam-sensitive nanoparticles that we observed before the first jump (180 min Fig. 1). Thus we agree with the reviewer that these particles are not necessarily ACP (nor can we be sure to be clusters) and therefore we will not mention ACP. Our TD model indeed did not consider ACP formation. The corresponding changes have been applied throughout the manuscript.

6) OCP and HA. The authors claim a two phase precipitation process. This has been reported in literature in many papers. Figure 1 is a strong evidence of it. Figure 3a/b show, maybe, even a third phase. The multiple-stage is a matter of fact. Nevertheless, the nature of the solid phases in each stage is not so clear. In my opinion the experimental evidences are not sufficient to support the statements about which phase is present at each stage. The most intense diffraction peak which clearly discriminate between OCP and HA occurs at too small angle to be detected by SAED (masked by the beam stop).

All the other peaks overlap and cannot be resolved by SAED. Moreover, many papers claim that OCP and HA coexist in a broad range of experimental conditions. Figure 2 (and the related SAED), unfortunately, does not help much to corroborate the author's speculations (here, I would exclude figure 2e/f for the reason of point 5). The experimental evidences show two morphologies. That is quite clear. But, in my opinion, not more than that. The small ED differences might be a consequence of the morphology as well as any other non-stoichiometric compounds in the continuous transformation between OCP and HA. Interestingly, in the experimental conditions studied, no ACP is detected (excluding the artifact due to drying). A double steps precipitation is often considered: first ACP and then a crystalline phase, according to the Oswald step rule. But it is not the case here, as also reported in [1]. The formation of DCPD cannot be excluded too (with acicular morphology). The first step seems consistent with the formation of a rather soluble solid phase (stage 190-250) followed by a second solid with lower solubility (280-350). From these almost linear sections, the solubility product can be calculated and compared with the literature values, in order to try to identify which phase might be present at that stage. Nevertheless, the so-calculated solubility values refer to the most soluble solid phase (whereas the diffraction patterns depict a volume averaged composition). The point here is that it does not matter which phase is produced: the value of the manuscript, in my opinion, is that there is a measurable difference between the nucleation conditions with or without an active surface. Therefore, I suggest to smooth the statement about the phase obtained because the presented evidence (in this manuscript as well as in many other papers on this topic), does not allow to line out clear statements (unless the diffraction peak at low angle, unique of the OCP structure, can be shown);

We identified the phases and reached the conclusions using a similar approach as in N. Sommerdijk's paper in Nat. Commun. in 2013 [4]. In fact, the characteristic (100) peak of OCP might not be able to help to distinguish the two phases neither:

“Only the (100) peak, which corresponds to planes with a large d-spacing of 1.85 nm along the thinnest dimension of the OCP plate-like morphology [5], was missing. This can be explained by the small thickness of the OCP-like plates in our samples (≈ 1.4 nm, see small-angle synchrotron X-ray scattering (SAXS) data Fig. 4b), which is less than the d-spacing between the (100) planes in OCP crystals. ”

However, we agree with the reviewer for the complexity in the Ca-P system and the need to smooth the statement in this manuscript as the presented evidence cannot serve as a full support to our current conclusions. Therefore, we have made corresponding changes in the manuscript, rather than focusing on the phase, the measurable difference between the nucleation conditions with and without an active surface is highlighted.

7) SI, chapter 8. Actually, already Figure 3 strongly support the findings reported in this chapter. For instance, in Fig 3c, an offset in the linear part Ca free curve is shown. By linear (range 25-125 min) extrapolation to time=0, a negative Ca free concentration can be measured, which should correspond to the amount of adsorbed Ca on the Ti NaOH surface. Doing the same exercise on the other curves of the same figure, it is clear that a very small amount of Ca is adsorbed in the case Ti NaOH HT and no adsorption occurs in case a and b. I suggest to compare the values obtained from Fig 3 and the values reported in table S4;

Thank you for pointing out these features which are consistent to our findings in SI-8. Indeed, in the case of Ti NaOH in Fig. 3c, the complexed & precipitated Ca given by the dotted line quickly rises to a certain value (between 5–10 mg/L), which is clearly different from the case of Ti and Ti HT. However, since the solution is constantly being diluted with the addition of calcium solution, the added Ca or expected free Ca^{2+} (assuming only complexation with phosphate and no precipitation) is not linear (the slope decreases over time). Thus a linear fit cannot accurately determine the true adsorbed value, for example a minor change in the slope value used can result in big change in the intercept. Therefore, we think the separate, quantitative adsorption experiments at different Ca concentrations presented in SI-8 is the appropriate method to determine the adsorption value. And the value we found at 6.5 mg/L in the case of Ti NaOH is similar to a rough estimate using the curve Fig. 3c. Nevertheless, as the reviewers said, the features from the four titration curves in Fig. 3 support our findings in SI-8, and an additional discussion is added to the SI.

8) SI, table S4; typo: Mes 1 \diamond Mes 2;

Thank you, this has been corrected.

9) SI, P9L29; Typo: Fig. S4 \diamond S6

Thank you, this has been corrected.

10) SI, Figure S12: “indicating a gradual nucleation”. This is conceptually incorrect. In this case, the process is just growth, not even heterogeneous nucleation; it is just continuous growth of HA on HA (as state on P23L11).

Thank you, this has been corrected.

11) General consideration: Figure 1 vs. Figure 3. In Figure 1 as well as Figure 3a and 3b, homogeneous nucleation occurs. Therefore, Ti and Ti HT are “not active”. In case of figure 3c and 3d, heterogeneous nucleation occurs instead. Therefore, a lower S is required to initiate the solid formation. Heterogeneous nucleation occurs because of an extra cohesion (or adhesion) energy contribution is provided during the nucleation stage on the Na-titanate layer. Thus, these surfaces are “active”, i.e. they show an apatite-forming ability. The solid formation on-set, define the “activity”: if it happens before the homogeneous threshold, the surface can be defined “active” since heterogeneous nucleation occurs. The fact that the precipitation is “less sharp” is a consequence of the different nucleation (and growth) rate, consequent of the lower saturation level (page 13 lines 5-9). The solid formed at lower S can be different; in fact, the equilibrium stage (e.g. 175-300 min, in fig, 3c/d) corresponds to a solid phase that have a solubility similar to the solid phase obtained at the end (275 min) of fig 3a/b. Therefore, in the presence of an active surface, the intermediated phase of fig 3a/b is not formed (in agreement with the author’s statement, P13L17-19). Because of heterogeneous nucleation, of the most thermodynamic stable phase is promoted. In my opinion, the concept of heterogeneous nucleation should be better introduced and discussed in the paper, since is the appropriate term which justifies the results. Instead, the section on P24L2-15, in my opinion, is rather misleading since (i) the influence of ACP (i.e. the artefact in the context of this manuscript) as well as (ii) pre-nucleation clusters (which are out of the scope of this

manuscript) are recalled and the (iii) heterogeneous nucleation concept (the key point of the manuscript) is considered not much relevant to assess the surface activity. In my opinion, the “material affinity to CaP” is “the ability to promoted heterogeneous nucleation”;

We agree with the reviewer that the concept of heterogeneous nucleation on active surfaces should be highlighted in the manuscript, thus major changes has been made on the original discussion section P24L2-15 as the reviewer pointed out. Corresponding changes are made throughout the manuscript.

12) I'm not able to evaluate the correctness of the in-vivo test. Nevertheless, the procedure and the explanation of the results are clearly described and convincing.

[1] <https://doi.org/10.1016/j.actbio.2018.05.027>

In conclusion, hoping that the authors find my criticisms constructive, I enjoyed the manuscript and I think is worthy of publication. Therefore, I would suggest elaborating the comments, smooth some statements which are not fully supported by experimental evidences or not relevant to support the major message of the manuscript; I would be happy to see a revised version.

Best regards.

Thank you very much for your recommendation and all the comments. We greatly appreciate the time you spent reviewing this manuscript and your detailed and constructive suggestions. These have helped us better interpret the results we had and improved the quality of the manuscript.

1. Carino, A., Ludwig, C., Cervellino, A., Müller, E. & Testino, A. Formation and transformation of calcium phosphate phases under biologically relevant conditions: Experiments and modelling. *Acta Biomater.* 74, 478–488 (2018).
2. Dey, A. et al. The role of prenucleation clusters in surface-induced calcium phosphate crystallization. *Nat. Mater.* 9, 1010–1014 (2010).
3. Bertram, A. K., Koop, T., Molina, L. T. & Molina, M. J. Clustering of Calcium Phosphate in the System $\text{CaCl}_2\text{-H}_3\text{PO}_4\text{-KCl-H}_2\text{O}$. *J. Phys. Chem. B. Phys. Chem. B* 103, 8230–8235 (1999).
4. Habraken, W. J. E. M. et al. Ion-association complexes unite classical and non-classical theories for the biomimetic nucleation of calcium phosphate. *Nat. Commun.* 4, 1507 (2013).
5. Terpstra, R. A. & Bennema, P. Crystal morphology of octacalcium phosphate: Theory and observation. *J. Cryst. Growth* 82, 416–426 (1987).

Reviewer #2 (Remarks to the Author):

It is an interesting and important topic to develop *in vitro* method to evaluate the *in vivo* performance of biomaterials. SBF method has been widely used because its ability to mimic the biological ion concentration and temperature, although it does have some limitations. In order to address these limitations, this study aimed to investigate a calcium titration method as a substitute for SBFs to evaluate the *in vivo* bioactivity. The paper was well written and organized, however the advantages of titration method was still debatable, compared to other *in vitro* methods such as some modified SBFs and the constant composition method developed by Dr. Nancollas. Some questions or comments are listed as follows.

1) The titration process could not mimic the biological condition that has relative stable ions strength and temperature. The different environments (such as Ca/P ratios, the presence of Mg^{2+} , and solution temperature...) may result in a different mineralization pathway for calcium phosphates. It might not be suitable to reflect the *in vivo* situation.

Due to the complexity of the human body condition, a complete simulation of the implantation site environment is difficult, especially if we also consider the complex reaction after implantation where initial inflammation reaction takes place that lowers the local pH [1] and the following bone remodeling process involving both osteoblast and osteoclast cells. In fact, what we know so far from previous studies conducted with chemically modified titanium and bioglass is that the material's ability to promote crystalline CaP (OCP/HA) formation is positively correlated to their *in vivo* bioactivity (bone-bonding ability). This is the reason that we propose to study this ability using a different approach, but still with controlled temperature and rather stable ionic strength (NaCl and Tris-HCl buffer added to both Ca solution and initial P solution). In addition, the titration procedure developed in this manuscript serves not just as one fixed protocol but rather a platform of study with the possibility of integrating other additional components in the system.

Certainly, the SBF method and the constant composition method developed by Dr. Nancollas are well-established methods working with one specific solution composition which can give us information about the induction period of CaP formation or the nucleation kinetics [2, 3]. However, we think the titration method is to some extent a generalized approach where the testing is no longer limited to a fixed solution composition but a dynamic varying solution compositions and thus is able to explore a broader range of possibilities. The other major advance here is the significant reduction in time needed for the tests while obtaining accurate predictions, which were further corroborated by the standard SBF test, cell culture tests and *in vivo* experiments.

An additional discussion is added to the manuscript with a reference of previous reports by Dr. Nancollas.

2) It is not clear whether this method is sensitive enough for other materials having moderate bioactivity, rather than the Ti-based materials which are well-studied bioactive materials.

The sensitivity of the current method can be tuned for different materials and surfaces by varying experimental parameters such as flow rate and solution concentrations. The high sensitivity of the calcium electrode, which can measure changes in calcium concentrations to <1 mg/L can help to detect minor features in the titration curve. Thus we have confidence in the titration experiment in better distinguishing different materials including ones with moderate to low activity, compared to the SBF method with a fixed solution composition.

Although the biocompatibility of titanium-based materials is well established and the *in vivo* bioactivity of chemically treated Ti has been observed in animal experiments [4], the NaOH treated titanium has not yet been used to replace the current Ti or hydroxyapatite coated Ti in practice. The major group of commercially available bioactive materials remains to be bioglass/biocement. These materials are widely used with confirmed bioactivity. Materials from the calcium phosphate family are also generally bioactive. HA itself is often used to coat Ti surface to achieve a better bone bonding. The effectiveness of our method in identifying the bioactivity of HA is experimentally confirmed and presented (Fig. S12). Besides these materials, other categories of bioactive materials are in fact rarely reported.

3) One big challenge of SBF is the inhibitory mineralization in the presence of proteins. It is not clear if same effect would happen when protein is in this titration system.

The inhibition effect is indeed a big challenge of SBF since chemically treated titanium with well-established *in vivo* bioactivity [4] was found to show no HA deposition when only 1 – 5 g/L of bovine serum albumin is added to the SBF, as we have demonstrated in 2017 [5]. In that paper, we found inhibitory effect of bovine serum albumin using the same four titanium surfaces as in this manuscript, thus we also believe in the titration system, a similar effect can be observed. Indeed, in the titration without the addition of any surfaces (homogeneous nucleation), our preliminary results showed a delay of ~60 min for the first precipitation event, although the two-step nucleation behavior remained unchanged. However, we think the studies involving proteins represent the next step following this manuscript which requires systematic studies using different titanium surfaces and protein concentrations. As we have mentioned earlier, we think this new titration protocol could be a powerful platform to study the effect of organic species, not only limited to proteins but also including various amino acids. While in the case of SBF, inhibitory effect can be observed simply due to lack of HA deposition, in the titration system, since a dynamic composition is used, nucleation event will eventually take place, thus providing the possibility to compare the effect of different organic species in a quantitative manner.

An additional brief discussion is added to the manuscript.

4) The decrease of free Ca^{2+} was measured to follow the mineralization process, but how to ensure that the Ca^{2+} drop was caused by the mineralization of surface, rather than the mineralization in the solution?

The Ca^{2+} drop can be a result of either homogeneous nucleation (mineralization in the solution) or heterogeneous nucleation (on the surface). Which way the nucleation happens depends on the bioactivity of the material. In the experiments, we are not trying to influence the way how the nucleation happens but to observe it and interpret from the results.

The presence of bio-inert material has little influence over CaP formation pathway or kinetics, thus homogeneous nucleation can happen, as in the case of Ti (or Ti HT) shown in the manuscript, many crystals were found in the solution unattached to any Ti surface, Fig. S7. This is consistent with the observation from the free calcium profile where no significant difference is observed compared to the control group (titration without external surfaces). The time needed for the nucleation is the same.

In the case of bioactive materials, as they are capable of inducing heterogeneous nucleation of CaP on their surfaces, the nucleation event takes place earlier. And the presence of CaP crystals on bioactive surfaces can be verified by TEM/SEM observations, as is carried out for the standard tests. This ability to promote heterogeneous nucleation of CaP is the key difference between a bioactive material and a bio-inert material.

5) It was noted in Fig.3 that the initial pH values were different in these groups. The groups of Ti with NaOH treatment have shown higher initial pH values. Would this higher pH affect the mineralization in the systems? Was the washing process (rinsed by DI water for 30s, 3 times) thorough enough to remove the NaOH residues from the sample surfaces?

pH is a very important factor in CaP precipitation system, which is the reason why we use Tris-HCl to buffer the solution pH. In practice, when we measure the solution pH, there are unfortunately slight but unpredictable drifts of the pH meter. Although we try to calibrate the pH meter before and after each experiment, this drift still results in ± 0.05 error. Due to this reason, we performed separate experiments in which we found that the pH change, upon addition of 0.1 g of any of the four titanium surfaces, are within 0.01, which is not significant. The results are mentioned in the legend of Fig. 4. Thus, the slight pH difference in the beginning of titration from Fig.3 is mostly from the system error. Nevertheless, the washing step is important to ensure the reproducibility of the results, as previously illustrated in the literature [6] and thus is strictly followed throughout all the experiments. From the many experiments carried out in our previous study [5] and in this study where our results are shown to be statistically significant, we believe that the washing routine used is sufficient.

In a broad sense, even though it is clear that from separate experiments, the addition of modified Ti powders in the Tris-HCl buffered phosphate solution results in very little pH change, the titration method can still be used to compare materials that can result in observable pH change or Ca depletion/release (as in the case of some bioglass) since the supersaturation value is used to compare the bioactivity, and the calculation of supersaturation already considers the pH and free Ca concentration, thus serving as a “fair” indicator for bioactivity.

6) The supersaturation degree towards OCP was calculated to indicate the bioactivity, however there is no direct evidence to prove the presence of OCP in Figure 3. Did the authors conduct the HR-TEM studies for the early formed crystals?

Yes, we did not include this due to repetition since in the case of Ti and Ti HT, firstly the free Ca profile resembles closely to the free Ca profile in titration experiment without external samples in terms of both shape (two characteristic peaks) and similar nucleation onset points. Secondly, many of the nucleation products (CaP crystals) were found unattached to Ti or Ti HT surfaces, an indication of homogeneous nucleation. Thirdly, their morphologies correspond well with the shape of OCP and HA in the homogeneous nucleation system (titration without samples). In the case of titration in the presence of Ti NaOH, results were given in Fig. S8, Fig. S9 and Fig. S10 with additional discussion.

Nevertheless, if we compare the CaP products from Ti experiment at the second peak (data originally not shown), CaP products from Ti NaOH experiment at first peak (same as Fig. S9–f), and HA nanocrystal (same as in Fig. 2–e), we can clearly see that the CaP products are still in the OCP region due to shift in the main peak at 3.6 nm^{-1} and extra middle component between two peaks as indicated by arrows compared to HA nanocrystals.

Fig. Normalized radial distribution density of the SAED patterns of needle-shaped CaP crystals from the titration experiment of Ti (second peak) and CaP crystals embedded in porous Ti NaOH surface from the titration experiment of Ti NaOH (first peak) plus a commercial HA nanocrystal analyzed using the same method.

In the end, we also agree with the comment from another reviewer that we should rather highlight the measurable difference between bioactive materials and bio-inert materials than focus on the crystalline phases since the determination of an exact crystalline phases with 100% proof in the complex Ca-P system might not be straightforward. Thus we have made corresponding changes throughout the manuscript and smoothed some of our statements.

We hope our answers to your comments are satisfactory. Thank you very much for your time reviewing this work. Your comments and recommendations are greatly appreciated.

1. Tsuchiya, H. Local anesthetic failure associated with inflammation: verification of the acidosis mechanism and the hypothetical participation of inflammatory peroxynitrite. *J. Inflamm. Res.* 41 (2008).
2. Wu, W. & Nancollas, G. H. Kinetics of Heterogeneous Nucleation of Calcium Phosphates on Anatase and Rutile Surfaces. *J. Colloid Interface Sci.* 199, 206–211 (1998).
3. Koutsoukos, P., Amjad, Z., Tomson, M. B. & Nancollas, G. H. Crystallization of calcium phosphates. A constant composition study. *J. Am. Chem. Soc.* 102, 1553–1557 (1980).
4. Nishiguchi, S. et al. Titanium metals form direct bonding to bone after alkali and heat treatments. *Biomaterials* 22, 2525–2533 (2001).
5. Zhao, W., Lemaître, J. & Bowen, P. A comparative study of simulated body fluids in the presence of proteins. *Acta Biomater.* 53, 506–514 (2017).
6. Uchida, M., Kim, H.-M., Kokubo, T., Fujibayashi, S. & Nakamura, T. Effect of water treatment on the apatite-forming ability of NaOH-treated titanium metal. *J. Biomed. Mater. Res.* 63, 522–30 (2002).

REVIEWERS' COMMENTS:

Reviewer #1 (Remarks to the Author):

Manuscript natcom_181192R2, "Rapid evaluation of bioactive Ti-based surfaces using an in-vitro titration method".

The paper has been revised, the new version is worth of publication.

Only minor suggestions:

- Section: discussion. Sub-chapter: Correlation between...; line 14: SI-17  SI-16
- Figure 4, caption: "... but not the minor ACP formation..." it reminds the previous version.

I would like also recommend a general remark, maybe useful to be mentioned in the manuscript: the general applicability of the method might be critically assessed:

1) The showed difference between the two kinds of surfaces is clear and the critical saturation level for heterogeneous nucleation is, most probably, a significant indicator of bioactivity. Nevertheless, heterogeneous nucleation depends on surface on which such events occur. Comparing the two series of samples (active and not active) the surface available for heterogeneous nucleation increase of at least 2 order of magnitude. In fact:

- For 0.1 g of Ti powder, 325-mesh, density 4.5 g cm⁻³ the overall surface can be estimated to be about 3.3 x 10⁻³ m².

- For the NaOH treated samples, if we consider a 2 micron layer of NaHTiOx, with density 2.5 g cm⁻³, packing factor 50%, deposited on the same surface of the untreated sample, the overall surface for heterogeneous nucleation is about 6.7 x 10⁻¹ m², therefore a factor about 200 larger than the untreated sample.

A generalized method should be normalized by surface, considering that the underlying process is surface-limited. In fact, let us assume that a sample coated with a dense layer of NaHTiO₃ might be available (i.e. no surface increase due to the treatment):

- would the electrodes be still sensitive enough to measure a curve similar to Figure 3c?

or,

- if Ti powders with a specific surface area larger by a factor 200 compared to those used (i.e., with a diam \approx 200 nm) are used, would the curve be similar to figure 3a?

2) In the experiments, Ti powders were used with an estimated overall surface of 3.3 x 10⁻³ m² and using 50ml of liquid. Now, if the surface bioactivity of an implant has to be measured, a similar surface should be tested. Thus, let us assume to evaluate the surface activity of implants similar to those used in the pre-clinical trials, as mentioned in the manuscript. It means that about 18 implants should be allocated in 50 mL of liquid with pH, ISE-Ca electrodes and dosing inlets. Not an easy task. In fact, it has not been done and Ti powders were used instead. Again, a generalized method should be normalized by real implant surface area available for heterogeneous nucleation (and reactor volume) otherwise false positive or false negative might occur (as by following the ISO23317 protocol).

Nevertheless, nowadays the ISO protocol suffers (at least) of the same limitations and the method presented in the manuscript is faster. Therefore, the claimed advantages are justified. Maybe the method is not yet at the level that can be considered as "the method" to assess bioactivity but it

represents an improvement.

Best regards.

A. Testino

Reviewer #2 (Remarks to the Author):

The authors supplemented sufficient discussions to make this work more complete and reliable. All the questions raised by reviewers have been addressed in the revised manuscript. The paper now shows without doubt that the calcium titration method proposed here has a great potential for evaluating the materials bioactivity. One suggestion is for the Keywords. "Sheep" was not the key for this work, probably "calcium titration" will be better words.

Reviewer #1 (Remarks to the Author):

Manuscript natcom_181192R2, "Rapid evaluation of bioactive Ti-based surfaces using an in-vitro titration method".

The paper has been revised, the new version is worth of publication. Only minor suggestions:

- Section: discussion. Sub-chapter: Correlation between...; line 14: SI-17  SI-16
- Figure 4, caption: "... but not the minor ACP formation..." it reminds the previous version.

These have been corrected.

I would like also recommend a general remark, maybe useful to be mentioned in the manuscript: the general applicability of the method might be critically assessed:

1) The showed difference between the two kinds of surfaces is clear and the critical saturation level for heterogeneous nucleation is, most probably, a significant indicator of bioactivity. Nevertheless, heterogeneous nucleation depends on surface on which such events occur. Comparing the two series of samples (active and not active) the surface available for heterogeneous nucleation increase of at least 2 order of magnitude. In fact:

- For 0.1 g of Ti powder, 325-mesh, density 4.5 g cm⁻³ the overall surface can be estimated to be about 3.3×10^{-3} m².

- For the NaOH treated samples, if we consider a 2 micron layer of NaHTiOx, with density 2.5 g cm⁻³, packing factor 50%, deposited on the same surface of the untreated sample, the overall surface for heterogeneous nucleation is about 6.7×10^{-1} m², therefore a factor about 200 larger than the untreated sample.

A generalized method should be normalized by surface, considering that the underlying process is surface-limited. In fact, let us assume that a sample coated with a dense layer of NaHTiO₃ might be available (i.e. no surface increase due to the treatment):

- would the electrodes be still sensitive enough to measure a curve similar to Figure 3c?

or,

- if Ti powders with a specific surface area larger by a factor 200 compared to those used (i.e., with a diam ≈ 200 nm) are used, would the curve be similar to figure 3a?

We started off our research using powders because this makes it easier to obtain a uniform mixing system and many bioactive materials such as bioglass is commercially available in the form of powders. Many new candidate materials are also often synthesized in the form of powders. Thus we aim to make it easier for people to use this protocol without having to worry about how to fix an implant inside the testing environment without introducing contamination and so on. Certainly, the use of powders also means increased surface area which facilitates the detection of heterogeneous nucleation event.

However, the surface area of several powders used in the experiments are not significantly different as the reviewer suggested. The experimentally measured values using nitrogen adsorption (BET model) showed that as-received Ti powder has a surface area of 0.15 m²/g and after heat treatment reduced to 0.12 m²/g. NaOH treated Ti (i.e. Ti NaOH) has a surface area of 2.4 m²/g and after heat treatment (Ti NaOH HT) reduced to 0.5 m²/g. The difference is certainly not 200 times as the reviewer suggested. If we compare Ti NaOH with Ti NaOH HT, there's a five-fold difference in surface area but the titration experiment showed no statistically significant difference between the two, which is in agreement with the results from *in vivo* sheep experiment.

Since the method does not aim to quantify the nucleation/growth kinetics but try to identify the nucleation onset point, normalization by surface area does not apply. In addition, the surface area of powders in the dispersed state in the liquid would differ from that measured in dry state using N₂ adsorption due to structural/morphological changes upon dispersion, making accurate estimates of surface area a practically challenging task.

2) In the experiments, Ti powders were used with an estimated overall surface of $3.3 \times 10^{-3} \text{ m}^2$ and using 50ml of liquid. Now, if the surface bioactivity of an implant has to be measured, a similar surface should be tested. Thus, let us assume to evaluate the surface activity of implants similar to those used in the pre-clinical trials, as mentioned in the manuscript. It means that about 18 implants should be allocated in 50 mL of liquid with pH, ISE-Ca electrodes and dosing inlets. Not an easy task. In fact, it has not been done and Ti powders were used instead. Again, a generalized method should be normalized by real implant surface area available for heterogeneous nucleation (and reactor volume) otherwise false positive or false negative might occur (as by following the ISO23317 protocol).

Nevertheless, nowadays the ISO protocol suffers (at least) of the same limitations and the method presented in the manuscript is faster. Therefore, the claimed advantages are justified. Maybe the method is not yet at the level that can be considered as “the method” to assess bioactivity but it represents an improvement.

One practical solution for experiments using samples with small surface area that are not available in a powder form is to reduce the titration rate. This practice would provide more time for the subtle nucleation events to develop (which is often slow at the beginning) before the addition of “too much” calcium ions, thus allowing a better detection of minor features in the free calcium profile. In this case, the method will take a little longer time but will still be significantly faster than the standard SBF experiment.

A short discussion is now added to the manuscript to point out the possible limitations and potential modifications of this method. We thank the reviewers for the time and effort spent reviewing this manuscript. The comments and suggestions are greatly appreciated.

Reviewer #2 (Remarks to the Author):

The authors supplemented sufficient discussions to make this work more complete and reliable. All the questions raised by reviewers have been addressed in the revised manuscript. The paper now shows without doubt that the calcium titration method proposed here has a great potential for evaluating the materials bioactivity. One suggestion is for the Keywords. “Sheep” was not the key for this work, probably “calcium titration” will be better words.

“Precipitation” is now used as key word replacing “Sheep”.